# Study on the Heterogeneous Nucleation Mechanism of SiC_p_/AZ91 Magnesium Matrix Composites under Pulse Current

**DOI:** 10.3390/ma16051993

**Published:** 2023-02-28

**Authors:** Xi Hao, Wei Liu, Teng Ma, Weixin Hao, Hua Hou

**Affiliations:** 1School of Materials Science and Engineering, Taiyuan University of Science and Technology, Taiyuan 030024, China; 2Collaborative Innovation Center of Ministry of Education and Shanxi Province, School of Materials Science and Engineering, North University of China, Taiyuan 030051, China; 3School of Mathematics, Jinzhong University, Jinzhong 030619, China

**Keywords:** pulse current, SiC_p_/AZ91 composites, microstructure refinement, heterogeneous nucleation

## Abstract

SiC_p_/AZ91D magnesium matrix composites with 30% SiC_p_ were successfully prepared by pulsed current melting in this work. Then, the influences of the pulse current on the microstructure, phase composition, and heterogeneous nucleation of the experimental materials were analyzed in detail. The results show that the grain size of both the solidification matrix structure and SiC reinforcement are refined by pulse current treatment, and the refining effect is gradually more obvious with an increase in the pulse current peak value. Moreover, the pulse current reduces the chemical potential of the reaction between SiC_p_ and Mg matrix, thus promoting the reaction between SiC_p_ and the alloy melt and stimulating the formation of Al_4_C_3_ along the grain boundaries. Furthermore, Al_4_C_3_ and MgO, as heterogeneous nucleation substrates, can induce heterogeneous nucleation and refine the solidification matrix structure. Finally, when increasing the peak value of the pulse current, the repulsive force between the particles increases while the agglomeration phenomenon is suppressed, which results in the dispersed distribution of SiC reinforcements.

## 1. Introdution

As a light structural material, magnesium (Mg) alloys exhibit good machinability, excellent castability, strong electromagnetic shielding ability, and good thermal conductivity and damping [1,2,3,4]. However, Mg alloys also have some defects, such as low strength, low modulus, and poor wear resistance and creep resistance, which seriously restricts its application in more industrial fields [5,6]. It is widely accepted that composite techniques adopting some suitable ceramic particles (i.e., SiO_2_, SiC and BN) into the Mg matrix can obtain good specific performance [7,8,9]. SiC-particle-reinforced magnesium matrix composites have many advantages, such as high specific strength, dimensional stability, low production cost, and so on, and their wear resistance and high temperature resistance have greatly improved. They have important applications in optical devices, nanotechnology, and nuclear and space materials science [10,11].

Excessive porosity and uneven particle distribution are two key issues in the operation of the stirring method for SiC-particle-reinforced magnesium matrix composites. Usually, hot extrusion is complicated in operation, and its cost is high, so it is not suitable for industrial production. The advantage of melt infiltration is that the volume fraction of the reinforcing phase in composites is not limited by its solid solubility in the metal matrix, but interface wetting between the matrix and the added phase is inevitable [12,13,14]. In recent years, many researchers have reported that the application of pulse currents to the manufacture of metal materials and the synthesis of ceramics could make the components of materials uniformly distributed, refine the materials’ organization, and improve their comprehensive properties [15,16,17,18]. For instance, Ma et al. [16] found that the solidification structure of a Cu–Pb monotectic alloy could be significantly improved by electric pulse treatment, and that the microstructure of the alloy could be refined. They also found that the Pb phase in Cu matrix could be uniformly distributed and that the solute interception effect was obvious. At the same time, Quan et al. [19] reported that the semi-solid microstructure of AZ91D alloy was changed by applying appropriate pulse current parameters, and equiaxed non-dendritic grains were formed instead of large grains with dendritic grains. However, until now, the application of pulse currents to the solidification process of composites has rarely occurred.

Therefore, in this work, SiC_p_/AZ91D magnesium matrix composites with a 30% volume fraction were successfully prepared by the stirring casting method under an electric pulse field. Meanwhile, the solidification microstructure evolution of SiC_p_/AZ91D under different pulse current parameters was studied, and the mechanisms of the pulse current in refining the solidification microstructure and promoting the uniform distribution of SiC_p_ were analyzed.

## 2. Materials and Methods

In this work, an SiC_p_/AZ91D magnesium matrix composite with an SiC_p_ content of 30 vol% was selected as the experimental object, and the matrix was an AZ91D magnesium alloy. The main components are shown in Table 1. The average particle size of SiC_p_ reinforcements is 30 μm.

### 2.1. SiC_p_/AZ91D Composites Prepared by the Semi-Solid Stirring Method

Firstly, AZ91D magnesium alloy is placed in a resistance furnace and heated to 750 °C. After the alloy is completely melted, the next step is cooling to a semi-solid state, starting the stirring paddle, and adding SiC_p_ preheated to 580 °C into the furnace. After stirring evenly, next is raising the temperature in the furnace to 700 °C; when the temperature reaches 700 °C, it is kept warm for ten minutes. At last, the SiC_p_/AZ91D composites are cast into a mold and molded at 150 MPa pressure to obtain SiC_p_/AZ91D composites.

### 2.2. Preparation of SiC_p_/AZ91D Composites under Pulse Current

Figure 1 shows a schematic diagram of the internal structure of a metal melt smelting device under the action of pulse current. Figure 2 shows the schematic diagram of the crucible size. Samples with a size of φ 14 mm × 10 mm were cut from the SiC_p_/AZ91D composite material; the oxide layer on the surface was polished with #2000 sandpaper and then put into the groove of a boron nitride crucible in the induction furnace after ultrasonic cleaning. The electrodes at both ends of the crucible are connected with the positive and negative electrodes of the high-power pulse power supply. When the inner pressure reaches 2 × 10^−4^ Pa, argon gas is added up to 50 kPa [20,21].

The SiC_p_/AZ91D composite specimens in the vacuum box are heated to 700 °C by a high-frequency induction heating device so that all the specimens in the crucible are melted. Subsequently, a pulse current is applied to the melt during cooling, and the action time is ten minutes. The temperature curve is shown in Figure 3. Because the oscillation, attenuation, and relaxation times of low-pulse-width pulse currents are very short, the influence of the Joule heating effect brought about by them can be ignored, so the low-pulse-width current parameter was selected in this experiment.

In order to compare the effects of different pulse currents on the solidification structure of the SiC_p_/AZ91D alloy, different pulse current parameters were applied to the SiC_p_/AZ91D alloy, as shown in Table 2.

### 2.3. Analysis and Detection Methods

The sample was cut into 10 mm-thick slices and polished for metallographic examination. An SS-550 scanning electron microscope (SEM) and XRD (diffraction of X-rays) (Bruker D8 AdvanceX-ray diffractometer) were used to observe the phase composition and for phase analysis of the SiC_p_/AZ91D composites. The microstructure of SiC_p_/AZ91D composites was characterized by TEM (transmission electron microscope) (jem-F200 field emission transmission electron microscope).

## 3. Results

Figure 4 is an SEM image showing the particle distribution of the magnesium matrix composites under a semi-solid stirring process. It reveals that SiC_p_/AZ91D magnesium matrix composites are composed of a dark matrix, AZ91D, and a light reinforcing phase, SiC. Additionally, the distribution of the SiC particles is relatively uniform, reflecting a good agglomeration phenomenon. Note that the size and distribution state of the SiC_p_ reinforcement phase are slightly different under different pulse current conditions. In order to characterize the improvement of the SiC_p_ distribution by the pulse current, the particle distribution was quantitatively characterized by the “grid method” [22]. Here, each sample photo is divided into 128 squares (magnification is 500 times), and the total number of particles is about 2400.

According to the total number of particles and the total number of squares, the average number of particles in each grid was calculated to be 4.7, which indicates that the more squares there are with 4–5 particles, the more uniform the distribution of SiC_p_ in the matrix. Figure 5 shows the statistical results for the SiC_p_ distribution under different pulse current conditions. It can be seen that the SiC distribution becomes more dispersed and uniform, and the segregation phenomenon weakens with the increase in the pulse current peak. Calculated by the Image Pro-Plus software, the average sizes of SiC under different pulse current conditions were identified and are displayed in Figure 6. It can be found that the size of SiC decreases with an increase in the pulse current peak. The above results show that the pulse current is beneficial to improve the distribution state of SiC.

Figure 7 shows the typical TEM microstructure morphology of an SiC_p_/AZ91D composites matrix under different pulse current peaks in the as-cast condition. As shown in Figure 7a, the α-Mg grains in the microstructure of SiC_p_/AZ91D composites are relatively coarse when no pulse current is applied. In Figure 7b, however, the α-Mg grains are refined to a certain extent under the action of 400 A current, and some grain sizes are reduced. Furthermore, in Figure 7c, when the peak current reaches 800 A, the α-Mg grains are further refined, but there are still larger grains in the solidification structure. Particularly, in Figure 7d, when the peak current reaches 1200 A, the grain size of the SiC_p_/AZ91 composites matrix is obviously improved, and the refinement effect on α-Mg is obvious. Figure 8 shows the grain size of α-Mg under different pulse current conditions. It can be seen that the pulse current has a significant effect reducing the grain size, and the effect is more obvious with an increase in the peak current.

Figure 9 shows the TEM morphology of the interface between SiC_p_ and the matrix at the peak pulse current of 1200 A. The microstructure morphology is shown in Figure 9a. In the matrix structure, Al_4_C_3_ is mixed with MgO, and Al_4_C_3_ is columnar. The morphology is similar to that in Figure 10. From the HREM analysis results in Figure 9b, Al_4_C_3_ and MgO appear at the interface between the SiC_p_ and the AZ91 matrix, without micropores, and there is a specific crystal orientation relationship. From Figure 9c,d, it is assumed that the chemical potential of SiC_p_ reacting with Mg and Al decreases due to the pulse current, and that Al_4_C_3_ and MgO are formed near the interface. The plane spacing of Al_4_C_3_ (110) is d=0.216 nm, and that of MgO is d=0.208 nm. The mixing effect of the two provides heterogeneous nucleation points for the composite. Compared with SiC_p_, Al_4_C_3_ can be used as a more excellent nucleation substrate, with better heterogeneous nucleation ability.

## 4. Discussion

Based on the above observations, it can be concluded that the solidification microstructure of SiC_p_/AZ91D was significantly affected by the application of a pulse current during solidification. Therefore, the effects of pulse currents on microstructure solidification behavior should be explained further.

### 4.1. Effects of Pulse Current on Grain Refinement of Mg Grain and SiC_p_

Figure 11 shows an XRD analysis of the SiC_p_/AZ91D composite test samples under different pulse currents. It can be seen that the diffraction peaks of α-Mg, β-Mg_17_Al_12_, and SiC_p_ decrease with an increase in pulse current peak value, which means that more α-Mg, β-Mg_17_Al_12_, and SiC_p_ participate in the reaction. The energy released by the pulse current provides external conditions for the reaction between SiC_p_ and α-Mg. Figure 6 shows that the size of SiC_p_ decreases under the action of the pulse current, indicating that some SiC_p_ reacts with Mg.

The effects of the pulse current on the chemical potential of reinforcer particles and the melt are as follows [23]:(1)$$\mu^{L}-\mu^{s}=\left(\mu_{o}^{L}-\mu_{o}^{S}\right)+FZ^{*}\left(g^{L}-g^{S}\right)$$
where *μ* is the electrochemical potential, *μ*_o_ is the normal chemical potential, *F* is the constant, *Z** is the charge on the metal particles, and *g* is the electrical potential difference between a point between phases and an infinite distance [23]. Assuming $g^{L}<g^{S}$, then:(2)$$\left(\mu^{L}-\mu^{s}\right)<\left(\mu_{o}^{L}-\mu_{o}^{S}\right)$$

The chemical potential of the reaction between SiC_p_ and α-Mg is reduced by the action of the pulse current, and the following reactions occur:
2Mg + SiC_p_ = Mg_2_Si + C(3)
4Al + 3C = Al_4_C_3_(4)

Therefore, Al_4_C_3_ can be seen in Figure 9. This has proved that carbide promoted the formation of a large number of carbon-containing crystal nuclei in the metal melt, which could lead to grain refinement [24,25]. In this work, Al_4_C_3_ and SiC_p_ provide heterogeneous nucleation numbers and induce a heterogeneous nucleation mechanism, thus refining the solidification structure. The extra undercooling of liquid metal is usually determined by the type and quantity of heterogeneous nuclei and the melt itself. Moreover, the affinity between SiC_p_ and α-Mg is much greater after applying the pulse current, so most of the initial SiC_p_ does not participate in the reaction, and they are in a cluster shape during the melt holding process. With an increase in the peak current of the pulse current, the agglomerated reinforced particles begin to gradually separate under the action of the external field and react with SiC and α-Mg in the melt, thus reducing the collision and coagulation between SiC_p_ after the reaction.

After the electric pulse acts on the alloy melt, the alternating electromagnetic force generated in the melt leads to the intensification of thermal convection, and the melt simultaneously obtains extra undercooling under the action of electromagnetic force [16]. Therefore, under the condition of high undercooling, the melt solidifies in a non-equilibrium way and a part of α-Mg begins to nucleate with SiC_p_ and Al_4_C_3_ as nucleation substrates, which results in the nucleus distribution being more uniform and the structure of α-Mg being finer.

Moreover, under the action of the alternating electromagnetic force, the magnetic contraction effect also plays a key role in refining the solidification structure of the magnesium matrix composites. Under the action of the magnetic field, the nuclei in the liquid melt will gather to the center, and the electromagnetic force will produce a certain electromagnetic pressure on the nuclei, which will lead to pressure produced by the magnetic field in the process of nucleus growth and inhibit the nucleus size from becoming larger. Due to the transient high-energy reciprocating action, pulse pressure and electromagnetic pressure lead to a reduction in nucleus size, and finally to α-Mg grain refinement.

### 4.2. Homogenization of SiC Reinforcement Phase by Pulse Current

Ren Jun et al. [26] pointed out that charging particles to the maximum extent was the technical key of the anti-agglomeration method. The necessary and sufficient condition is that the particles will not agglomerate under the action of the pulse current. The total force, *F_T_,* between particles is repulsive and can be expressed as follows:(5)$$F_{T}=F_{W}+F_{qc}+F_{ek}\;(F_{T}>0)$$
where *F_W_* is the Van der Waals force, *F_qc_* is the magnetic attraction force of the particles, and *F_ek_* is the Coulomb repulsive force between two particles. As long as the electrostatic repulsive force between particles is greater than the attraction force caused by agglomeration, the particles will be in a better-dispersed state.

The van der Waals force *F_W_* can be expressed as [26]:(6)$$F_{W}=-\frac{Ad}{24H^{2}}$$
where *A* is the Hamaker constant of the particles, which is related to the surface of the particles; *H* is the distance between the particles; and *D* denotes the diameter of the particles.

The coulomb repulsive electrostatic force *F_ek_* is expressed as:(7)$$F_{ek}=\frac{1}{4\pi \varepsilon_{0}}\cdot \frac{q_{1}q_{2}}{r^{2}}$$
where *r = d + H*.

Assuming that the electrical quantities of the two particles are the same, there are:(8)$$q_{1}=q_{2}=1+\left[2\left(\frac{\varepsilon_{r}-1}{\varepsilon_{r}+1}\right)\right]\pi \varepsilon_{0}d^{2}E_{0}$$

The Coulomb repulsion force/between two particles is:(9)$$F_{ek}=\frac{1}{4}{\left(\frac{3\varepsilon_{r}-1}{\varepsilon_{r}+1}\right)}^{2}\pi \varepsilon_{0}E_{0}^{2}\cdot \left(\frac{d^{2}}{d+H}\right)$$
where *ε*_0_ is the dielectric constant in vacuum; *ε_r_* is the relative dielectric constant of the particles; *E*_0_ represents the electric field strength; *d* is the diameter of the reinforcing particles; and *H* is the distance between the particles. When the particles segregate, the particle spacing is much smaller than the particle radius and *H* << *d* is obtained, then Equation (5) is as follows:(10)$$F_{ek}=\frac{1}{4}{\left(\frac{3\varepsilon_{r}-1}{\varepsilon_{r}+1}\right)}^{2}\pi \varepsilon_{0}Ed^{2}$$

The reinforcements selected in this paper are SiC_p_ with a diameter of 30 μm. It can be seen from Equation (6) that, with an increase in pulsed electric field intensity, the repulsive force between particles increases and the agglomeration phenomenon weakens. Increasing electric field intensity is beneficial to the dispersion distribution of SiC_p_ particle reinforcements.

## 5. Conclusions

The heterogeneous nucleation of the solidification structure of SiC_p_/AZ91D magnetic matrix composites under a pulse current is analyzed in this work. The main results are as follows:(1)The microstructure of SiC_p_/AZ91D magnesium matrix composites is significantly refined by using a pulse current during solidification. In addition, the segregation of SiC_p_ particles is effectively inhibited, and this effect becomes more and more obvious with the increase in pulse current peak value.(2)Al_4_C_3_ can be used as a more effective nucleation substrate due to its appearance near the interface between MgO and SiC_p_, and the size of SiC_p_ is significantly reduced by applying a pulse current. Meanwhile, the pulse current reduces the chemical potential of the reaction between SiC_p_ and the matrix, promotes the reaction between them, and produces Al_4_C_3_ near the interface. Interestingly, both Al_4_C_3_ and MgO can induce heterogeneous nucleation and refine the solidification structure, which is because they can be regarded as an effective heterogenous nucleation substrate.(3)A pulse current can significantly improve the segregation of SiC_p_ magnesium matrix composites. Additionally, the repulsive force between SiC_p_ particles increases and the agglomeration phenomenon weakens with increasing pulsed electric field intensity. The application of a pulsed current is beneficial to the dispersion distribution of SiC_p_ particles in the matrix.

## Figures and Tables

**Figure 1 materials-16-01993-f001:**
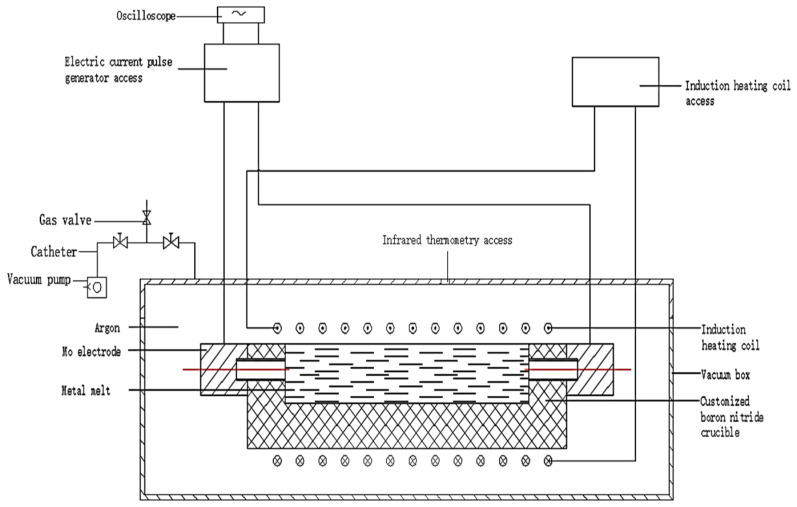
Melt smelting device under pulse current.

**Figure 2 materials-16-01993-f002:**
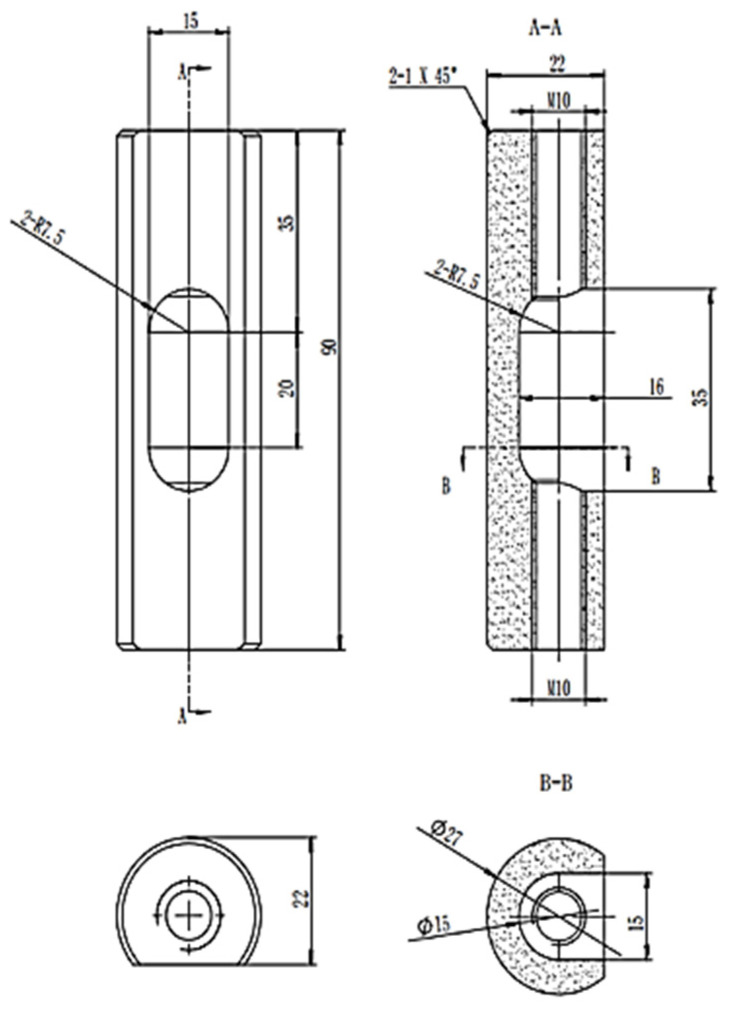
Crucible dimensions of the melt smelting device under pulse current.

**Figure 3 materials-16-01993-f003:**
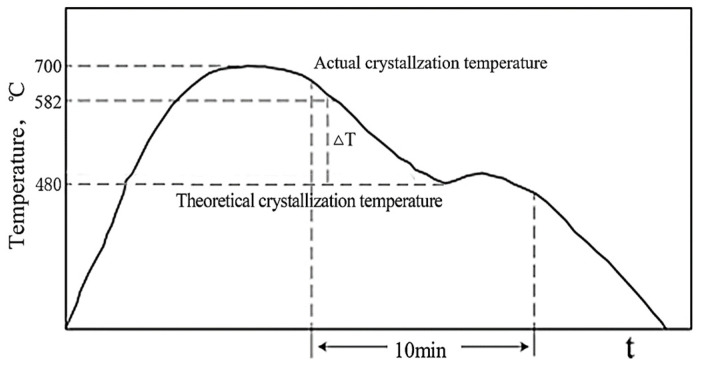
Schematic diagram of the action time of the ECP on the molten metal.

**Figure 4 materials-16-01993-f004:**
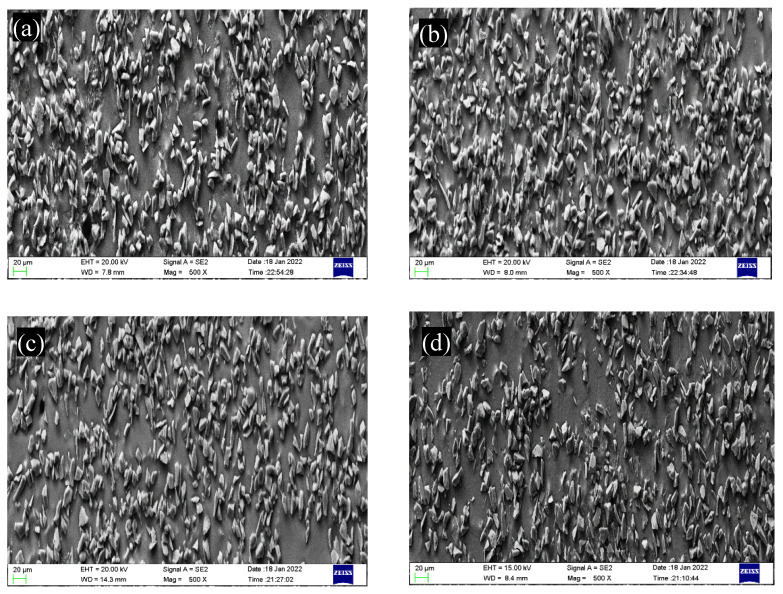
SEM images showing 500 times SiC_p_/AZ91D composite particle distribution. (**a**) 0 A; (**b**) 400 A; (**c**) 800 A; (**d**) 1200 A.

**Figure 5 materials-16-01993-f005:**
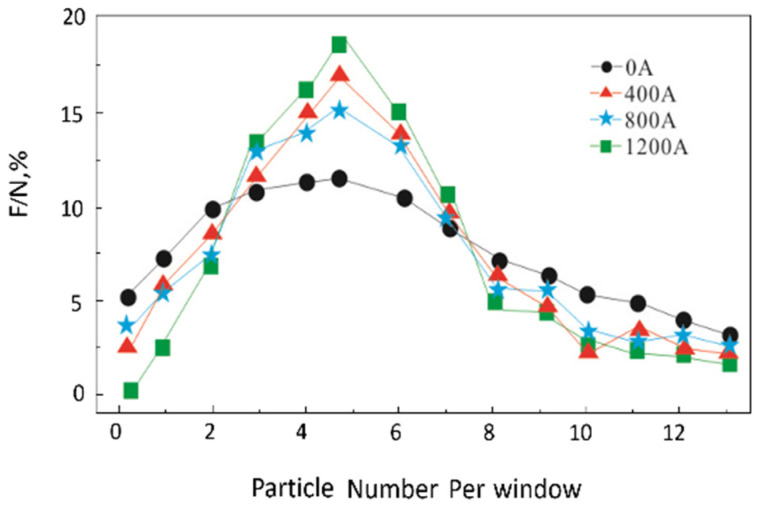
Grain size distribution of SiC under different pulse current conditions.

**Figure 6 materials-16-01993-f006:**
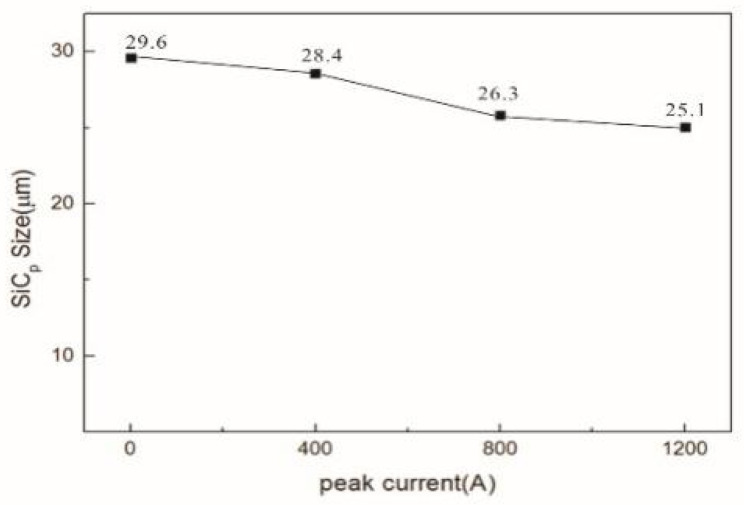
Average grain size of SiC under different pulse current conditions.

**Figure 7 materials-16-01993-f007:**
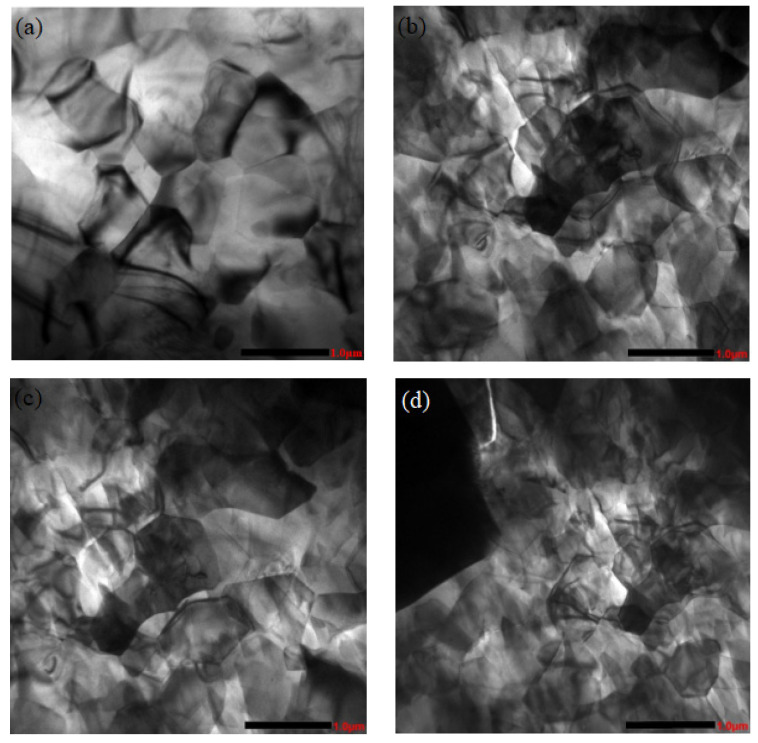
TEM morphologies of the SiC_p_/AZ91D matrix under different pulse current peaks. (**a**) 0 A (**b**) 400 A (**c**) 800 A (**d**) 1200 A.

**Figure 8 materials-16-01993-f008:**
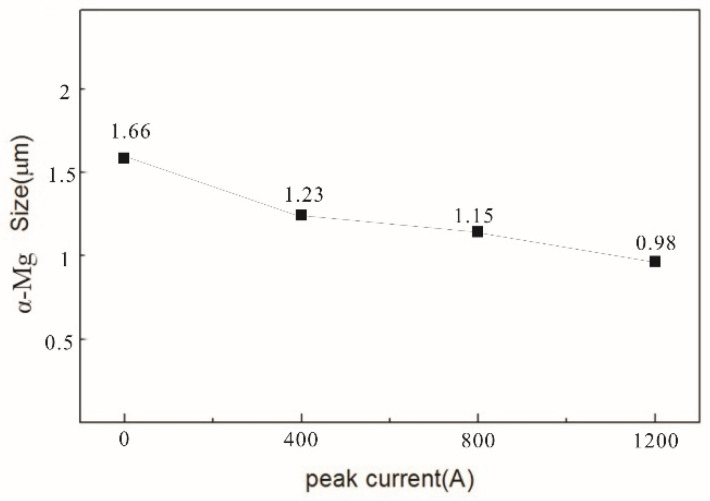
SiC_p_/AZ91: average grain size of the composites.

**Figure 9 materials-16-01993-f009:**
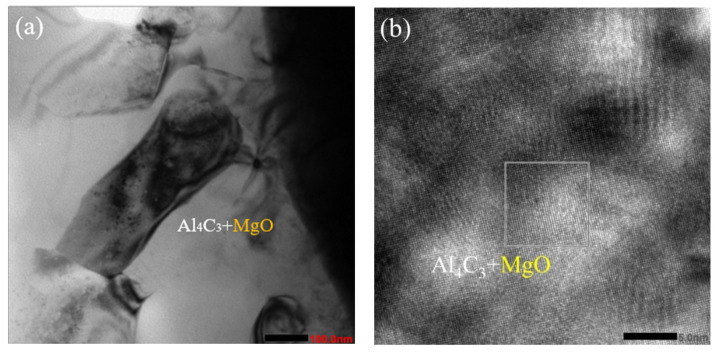
SiC_p_/AZ91: TEM microstructure near the interface of composites. (**a**) morphology of the matrix structure, (**b**) HREM image of the matrix structure (**c**) Calibrate HREM image (**d**) filtration image of (**c**).

**Figure 10 materials-16-01993-f010:**
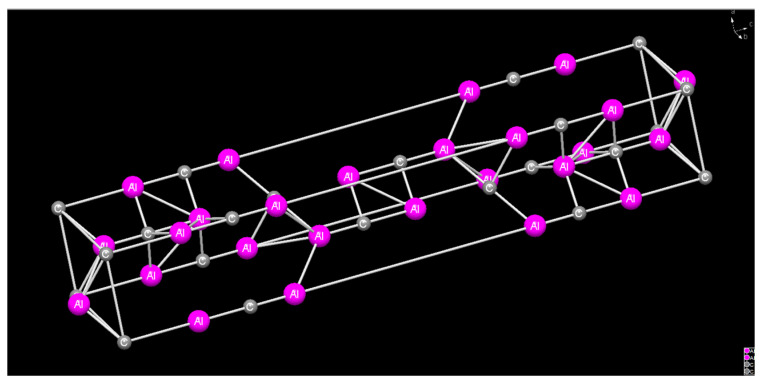
Al_4_C_3_crystal structure.

**Figure 11 materials-16-01993-f011:**
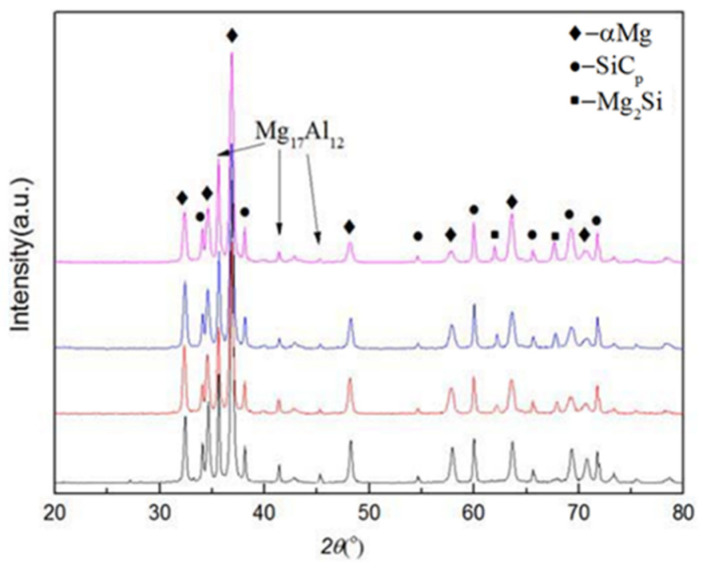
XRD diffraction pattern of SiC_p_/AZ91D composites.

**Table 1 materials-16-01993-t001:** Chemical composition of AZ91D alloy [9].

Wt%					
Mg	Al	Zn	Mn	Si	Cu
90.015	9	0.67	0.25	0.05	0.015

**Table 2 materials-16-01993-t002:** Parameters of the pulsed current in this work.

Sample	Current Peak, A	Voltage, V	Current Frequency, Hz	Pulse Width, µs
a	0	0	0	0
b	400	50	1000	20
c	800	150	1000	20
d	1200	200	1000	20

## Data Availability

The data that support the findings of this study are available from the corresponding author upon reasonable request.

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
