# Peer review of "Study on the Heterogeneous Nucleation Mechanism of SiCp/AZ91 Magnesium Matrix Composites under Pulse Current"

_materials, 2023, doi:10.3390/ma16051993_

Round 1

Reviewer 1 Report

Referee report on “Study on Heterogeneous Nucleation Mechanism of SiCp/AZ91 Magnesium Matrix Composites under Pulse Current

Although this topic is of some interest, this manuscript in its present form cannot be recommended for publication and requires some improvement and clarification.

1.     Abstract.  Both SiCp/AZ91D must be explained.

2.     Line 33. What crystalline modification of silicon carbide is meant.

3.     Furthermore, the introduction needs more general information about SiC and its important applications in optical devices, nanotechnology and nuclear and space material science. This is important to attract more reader interest and further incentive applications. For some of them, see, for example:

a)     Huczko, A., Dąbrowska, A., et al . Silicon carbide nanowires: synthesis and cathodoluminescence. physica status solidi (b), 2009, 246(11‐12), 2806-2808.

b)     Ning G., Zhang L.,  et al Damage and annealing behavior in neutron-irradiated SiC used as a post-irradiation temperature monitor

(2022) Nuclear Instruments and Methods in Physics Research, Section B: Beam Interactions with Materials and Atoms, 512, pp. 91-95.

4.     Line 43-45 and line 53.  Note that it has recently been demonstrated that pulsed electron flows are effectively useful not only for metals, but also for the synthesis of ceramics.  See, for example:

Karipbayev, Z. T., Lisitsyn, V. M., et al (2020). Time-resolved luminescence of YAG: Ce and YAGG: Ce ceramics prepared by electron beam assisted synthesis. Nuclear Instruments and Methods in Physics Research Section B: Beam Interactions with Materials and Atoms479, 222-228.

5.     Table 1. First line of the Table needs proper description.  Please also complete this Table by error bars.

6.     Fig. 1 and 2 need better quality.

7.     What determines the time of 10 minutes?

8.     Results.  What crystal modification do SiC have?

9.     Fig. 5. The legend at the bottom of the figures is almost invisible.

10.  Fig. 9.  The presented data need error bars.

11.   References: [4] No volume or page numbers.

In general, the manuscript is interesting and can be considered for publication after constructive reflection on the above comments.

Author Response

Response to Reviewer 1 Comments

Thank you for your time and wonderful suggestions to my manuscript. I have revised
my manuscript according to your comments as following :

Point 1: Abstract.  Both SiCp/AZ91D must be explained.

Response 1: Thank you for your advice. According to your advice, It has been revised.

Point 2: Line 33. What crystalline modification of silicon carbide is meant.

Response 2: Thank you for your advice. Line 33 stated that SiC as a reinforcement should be used in magnesium matrix composites. The paper is devoted to studying the change of SiC size caused by pulsed electric field

Point 3: Furthermore, the introduction needs more general information about SiC and its important applications in optical devices, nanotechnology and nuclear and space material science. This is important to attract more reader interest and further incentive applications. For some of them, see, for example:

  1. a)     Huczko, A., Dąbrowska, A., et al . Silicon carbide nanowires: synthesis and cathodoluminescence. physica status solidi (b), 2009, 246(11‐12), 2806-2808.
  2. b)     Ning G., Zhang L.,  et al Damage and annealing behavior in neutron-irradiated SiC used as a post-irradiation temperature monitor

(2022) Nuclear Instruments and Methods in Physics Research, Section B: Beam Interactions with Materials and Atoms, 512, pp. 91-95.

Response 3: Good suggestion. According to your advice, I have cited the article.

Point 4:   Line 43-45 and line 53.  Note that it has recently been demonstrated that pulsed electron flows are effectively useful not only for metals, but also for the synthesis of ceramics.  See, for example:

Karipbayev, Z. T., Lisitsyn, V. M., et al (2020). Time-resolved luminescence of YAG: Ce and YAGG: Ce ceramics prepared by electron beam assisted synthesis. Nuclear Instruments and Methods in Physics Research Section B: Beam Interactions with Materials and Atoms, 479, 222-228.

Response 4: Good suggestion. According to your advice, it has been revised.

Point 5: Table 1. First line of the Table needs proper description.  Please also complete this Table by error bars.

Response 5: Good suggestion. I’m so sorry for my carelessness. I have been revised the Table 1.

Point 6:   Fig. 1 and 2 need better quality.

Response 6: Thank you for your advice. According to your advice, It has been revised.

Point 7: What determines the time of 10 minutes?

Response 7: Good suggestion. In this experiment, the electric pulse applied to the material in the molten state is proved by many experiments, so that the pulse current can fully act on the melt in the crucible and improve the structure. The following can be used as arguments:

1https://kns.cnki.net/kcms2/article/abstract?v=kxaUMs6x7-4I2jr5WTdXti3zQ9F92xu00JiK22IWDK_wNv94U4IP9PvIkBiXCAdDNSVItXPkAtORnKhEXT61YtqQIbdMLVIr&uniplatform=NZKPT

2Effects of High-Density Pulse Currents on the Solidification Structures of SiCp/AZ91D Composites[J]. Advances in Materials Science and Engineering, 2019, 2019.

Point 8:  Results.  What crystal modification do SiC have?

Response 8: Good suggestion. The size and distribution state of SiCp reinforcement phase are slightly different under different pulse current conditions.

Point 9:  Fig. 5. The legend at the bottom of the figures is almost invisible.

Response 9: Thank you for your advice. According to your advice, it has been revised.

Point 10:   Fig. 9.  The presented data need error bars.

Response 9: Thank you for your advice. Fig. 9 shows that both Al4C3 and MgO can induce heterogeneous nucleation and refine solidification structure under the action of pulse current, which is due to they can be re-garded as effective heterogenous nucleation substrate. For the error bars, I will continue to pay attention to it in the follow-up study.

Point 11:  References: [4] No volume or page numbers.

Response 9: Thank you for your advice. According to your advice, it has been revised.

Thank you very much for your time and help. I am looking forward to hear from you
soon.
With best wishes,
Yours sincerely,
Hua Hou
2023-02-20

Reviewer 2 Report

Review for 

materials-2230063

 Study on Heterogeneous Nucleation Mechanism of SiCp/AZ91 Magnesium Matrix Composites under Pulse Current

The application of Mg alloys is limited by many aspects. The authors trying to solve the problems of porosity and uneven particle distribution during the process of SiC reinforcement. I believe, this work is within the scope of this journal and recommend it for publication after some revision. Albeit this work is in good shape, I think, nonetheless, that the manuscript could be improved if the authors could address the comments and recommendations I listed below.

I'm highly curious about the corrosion resistance of your alloy. 

 The novelty of your research should be highlighted in the abstract. 

I notice you use SiCp in your study. What does the meaning of "p" in SiCp?

You may need to have a better background study of Mg and the strengthening mechanism of Si. You can cite the following article:

Study on Strengthening and Toughening of Mechanical Properties of Mg-Li Alloy by Adding Non-Rare-Earth Elements Al and Si. JOM 74, 2554–2565 (2022). https://doi.org/10.1007/s11837-022-05296-y

Line 65 table 1: What does the "wB" mean? The weight percentage of atomic percentage?

Line 121: More set-up parameters of SEM and XRD are needed. Like applied voltage and a working distance of SEM. The energy source of XRD. 

Figure 5: Which figure is a, b, c, and d? You should label it in your 4 figures.

How you measured the grain size distribution of SiC?

Line 228 Figure 12: You may need to add the XRD PDF card of each phase or cite other people's work. 

Line 239: Is there any source of your equation (1). Give a citation to it.

Significance :

. The scientific content of this paper is correct. 

. The technical quality of this paper is correct. 

Scientific soundness :

. The subject addressed in this paper is relevant. 

Interest to the readers :

. In my opinion, the method of this paper seems to be interesting for the readership of the journal.  

Overall, this review is in good shape. You may consider my above suggestions. 

Author Response

Response to Reviewer 2 Comments

Thank you for your time and wonderful suggestions to my manuscript. I have revised
my manuscript according to your comments as following :

Point 1: I'm highly curious about the corrosion resistance of your alloy.

Response 1: Good suggestion. In the follow-up study, I will study it in depth.

Point 2: I notice you use SiCp in your study. What does the meaning of "p" in SiCp?

Response 2: Thank you for your advice. According to your advice, "p" in SiCp is powder.  

Point 3:You may need to have a better background study of Mg and the strengthening mechanism of Si. You can cite the following article:

Study on Strengthening and Toughening of Mechanical Properties of Mg-Li Alloy by Adding Non-Rare-Earth Elements Al and Si. JOM 74, 2554–2565 (2022). https://doi.org/10.1007/s11837-022-05296-y

Response 3: Good suggestion. I have cited the article.

Point 4: Line 65 table 1: What does the "wB" mean? The weight percentage of atomic percentage?

Response 4: Good suggestion. I’m so sorry for my carelessness. I have been revised the Table 1.

Point 5: More set-up parameters of SEM and XRD are needed. Like applied voltage and a working distance of SEM. The energy source of XRD. 

Response 5: Thank you for your advice. The parameters of SEM in bottom of Fig 4.

The test scope of XRD is 20°~80 °,Scanning speed is 1°/min.

Point 6: Figure 5: Which figure is a, b, c, and d? You should label it in your 4 figures.

Response 6: Thank you for your advice. According to your advice, it has been revised.

Point 7: How you measured the grain size distribution of SiC?

Response 7: Good suggestion. In order to characterize the improvement of SiCp distribution by pulse current, the particle distribution was quantitatively characterized by "grid method" [21]. Here, each sample photo is divided into 128 squares (magnification is 500 times), and the total number of particles is about 2400.

Point 8: Line 228 Figure 12: You may need to add the XRD PDF card of each phase or cite other people's work.

Response 8: Good suggestion. This paper describes that Heterogeneous Nucleation Mechanism of SiCp/AZ91 Magnesium Matrix Composites under Pulse Current. From formula (3) and (4)

can be seen that the diffraction peaks of α- Mg, β- Mg17Al12 and SiCp decrease with the increase of pulse current peak value, which means that more α- Mg, β- Mg17Al12 and SiCp particulate in the reaction. The reduction of SiCp size (Fig. 5) and The reduction of SiCp size (Fig. 5) and the appearance of Al4C3 (Fig. 11) both verified the reaction. Fig. 12 is obtained by processing the original data in JUDE6. I will continue to pay attention to PDF card in the follow-up researchers

Point 9: Line 239: Is there any source of your equation (1). Give a citation to it.

Response 9: Thank you for your advice,it has been revised.

Thank you very much for your time and help. I am looking forward to hear from you
soon.
With best wishes,
Yours sincerely,
Hua Hou
2023-02-20

Reviewer 3 Report

Dear Authors,

I have some remarks for your paper:

1.  I think that you should improve references. Most of references are more than 10 yares old. 

2. You should add the references with yours papers. You have experiences in applying pulse current for melting/remelitng.

3. You are using SiCp and  SiCp with subscript at the same time. Use one of them.

4. Part of references in text are noted in supersrcipts e.g. [16]

4. Line 63 - I think that information about range of size would be more interesting. And the change of particles size after treatment would be important information and its effect on the properties.

5. Table 1. Explain "ωB." 

6. Figure 1  - you should explain the notes 1-10 as in other your publications.

7. Line 127  - There is not fig.4 in paper. Fig. 5 - there should figure 4. 

8. Line 261 - there is strikethrough text.

Author Response

Response to Reviewer 3 Comments

Thank you for your time and wonderful suggestions to my manuscript. I have revised
my manuscript according to your comments as following :

Point 1:  I think that you should improve references. Most of references are more than 10 yares old. 

Response 1: Good suggestion. I have been revised. In recent years, there are few references about the effect of pulse current on liquid metal, especially in magnesium matrix composites. I will continue to pay attention to the references.

Point 2: You should add the references with yours papers. You have experiences in applying pulse current for melting/remelitng.

Response 2: Thank you for your advice. According to your advice, References [17] and [18] have added.

Point 3: You are using SiCp and SiCp with subscript at the same time. Use one of them.

Response 3: Good suggestion. I’m so sorry for my carelessness. According to your advice, It has

fixed.

Point 4: Part of references in text are noted in supersrcipts e.g. [16]

Response 4: Good suggestion. I’m so sorry for my carelessness. I have been revised.

Point 5: Line 63 - I think that information about range of size would be more interesting. And the change of particles size after treatment would be important information and its effect on the properties.

Response 5: Thank you for your advice. In the follow-up study, I will continue to study the specific changes of particle size

Point 6:  Table 1. Explain "ωB."

Response 6: Thank you for your advice. According to your advice, It has been revised.

Point 7: Figure 1  - you should explain the notes 1-10 as in other your publications.

Response 7: Good suggestion. According to your advice, Fig.1 has been revised. The pulse current is applied to the melt during cooling, and the action time is ten minutes.

Point 8:  Line 127  - There is not fig.4 in paper. Fig. 5 - there should figure 4. 

Response 8: Good suggestion. I’m so sorry for my carelessness. I have been revised.

Point 9: Line 261 - there is strikethrough text.

Response 9: Thank you for your advice. The size of the marker is provided at the bottom right which is 1μm.

Thank you very much for your time and help. I am looking forward to hear from you
soon.
With best wishes,
Yours sincerely,
Hua Hou
2023-02-20

Reviewer 4 Report

1. Please remove keyword "keyword".
2. Please check the subscript symbols for the compounds. For ex. SiO2 on line 32 (2 must be subscript symbol).
3. Line 61: AZ91D, not AZ9D. Please fix.
4. What in the Table 1 ωB. means? And in what percent (wt. or at.) the composition of alloy is provided?
5. Line 69: What the temperature of melt cooling for obtaining semi-solid state?
6. Line 73: MPa, not Mpa. Please check.
7. Line 75: Metal melt smelting device is confusing.
8. Figure 3: Which of the temperatures are the theoretical solidification temperature? The term "crystallization" is inappropriate here.
9. Figure 8: The size of the marker is not provided.
10. Line 258: The "p" in SiCp is crossed out, why? The same is true for "at the same time" at line 261.

5. 

Author Response

Response to Reviewer 4 Comments

Point 1: Please remove keyword "keyword".

Response 1: Good suggestion. I’m so sorry for my carelessness. According to your advice, It has

been revised.

Point 2: Please check the subscript symbols for the compounds. For ex. SiO2 on line 32 (2 must be subscript symbol).

Response 2: Thank you for your advice. According to your advice, It has been revised.

Point 3: Line 61: AZ91D, not AZ9D. Please fix.

Response 3: Good suggestion. I’m so sorry for my carelessness. According to your advice, It has

been fixed.

Point 4: What in the Table 1 ωB. means? And in what percent (wt. or at.) the composition of alloy is provided?

Response 4: Good suggestion. I’m so sorry for my carelessness. I have been revised the Table 1.

Point 5: Line 69: What the temperature of melt cooling for obtaining semi-solid state?

Response 5: Thank you for your advice. The melt cooling for obtaining semi-solid state at 580 ℃. “After the alloy is completely melted, cooling to semi-solid state, starting the stirring paddle, and adding SiCp preheated at 580 ℃ into the furnace.” Three cases are conducted at the same temperature.

Point 6: MPa, not Mpa. Please check.

Response 6: Thank you for your advice. According to your advice, It has been revised.

Point 7: Line 75: Metal melt smelting device is confusing.

Response 7: Good suggestion. According to your advice, Fig.1 has been revised. The pulse current is applied to the melt during cooling, and the action time is ten minutes.

Point 8: Figure 3: Which of the temperatures are the theoretical solidification temperature? The term "crystallization" is inappropriate here.

Response 8: Good suggestion. According to your advice, Fig. 3 has been revised. 480 ℃ is the theoretical crystallization. In fact, there is a difference between the actual crystallization temperature and the theoretical crystallization temperature, which is called undercooling. The crystallization temperature refers to the initial temperature of the melt from liquid phase to solid phase, and the initial temperature of the solidification process, so it is more appropriate to use crystallization temperature here.

Point 9: Figure 8: The size of the marker is not provided.

Response 9: Thank you for your advice. The size of the marker is provided at the bottom right which is 1μm.

Point 10:   Line 258: The "p" in SiCp is crossed out, why? The same is true for "at the same time" at line 261.

Response 10: Good suggestion. I’m so sorry for my carelessness. I have been revised.

Round 2

Reviewer 1 Report

After quite detailed and comprehensive revision, this manuscript can be recommended for publication.

Reviewer 2 Report

After checking the draft of the response to the comments, and the corresponding revisions in the revised manuscript, I found that the authors have accomplished the recommended revision to address all my concerns.